# Animal Welfare Management in a Digital World

**DOI:** 10.3390/ani10101779

**Published:** 2020-10-01

**Authors:** Henry Buller, Harry Blokhuis, Kees Lokhorst, Mathieu Silberberg, Isabelle Veissier

**Affiliations:** 1Department of Geography, University of Exeter, Rennes Drive, Exeter EX4 4RJ, UK; 2Department of Animal Environment and Health, Swedish University of Agricultural Sciences, P.O. Box 7068, 750 07 Uppsala, Sweden; harry.blokhuis@slu.se; 3Wageningen UR, Wageningen Livestock Research, P.O. Box 338, 6700AH Wageningen, The Netherlands; kees.lokhorst@wur.nl; 4UMR Herbivores, Université Clermont Auvergne, INRAE, VetAgro Sup, 63122 Saint-Genès-Champanelle, France; mathieu.silberberg@inrae.fr (M.S.); isabelle.veissier@inrae.fr (I.V.)

**Keywords:** animal welfare, precision livestock farming, welfare management, welfare monitoring, welfare assurance

## Abstract

**Simple Summary:**

The digital revolution opens possibilities to use multiple sensors, a data infrastructure and data analytics to monitor animals or their environment 24/7. Precision Livestock Farming (PLF) offers significant opportunities for a holistic, evidence-based approach to the monitoring and surveillance of farmed animal welfare. To date, the emphasis of PLF has been on animal health and productivity. If PLF develops further along these lines, there is a risk that animal health and productivity define welfare. A combined multi-actor approach that brings together industry, scientists, food chain actors, policy-makers and NGOs is needed to develop and use the promise of PLF for the creative and effective improvement of farmed animal welfare, not only taking into account their physical welfare but also their mental one.

**Abstract:**

Although there now exists a wide range of policies, instruments and regulations, in Europe and increasingly beyond, to improve and safeguard the welfare of farmed animals, there remain persistent and significant welfare issues in virtually all types of animal production systems ranging from high prevalence of lameness to limited possibilities to express natural behaviours. Protocols and indicators, such as those provided by Welfare Quality, mean that animal welfare can nowadays be regularly measured and surveyed at the farm level. However, the digital revolution in agriculture opens possibilities to quantify animal welfare using multiple sensors and data analytics. This allows daily monitoring of animal welfare at the group and individual animal level, for example, by measuring changes in behaviour patterns or physiological parameters. The present paper explores the potential for developing innovations in digital technologies to improve the management of animal welfare at the farm, during transport or at slaughter. We conclude that the innovations in Precision Livestock Farming (PLF) offer significant opportunities for a more holistic, evidence-based approach to the monitoring and surveillance of farmed animal welfare. To date, the emphasis in much PLF technologies has been on animal health and productivity. This paper argues that this emphasis should not come to define welfare. What is now needed is a coming together of industry, scientists, food chain actors, policy-makers and NGOs to develop and use the promise of PLF for the creative and effective improvement of farmed animal welfare.

## 1. Introduction

Modern livestock farming has long been predicated upon ever-increasing levels of mechanisation. As herd and flock sizes increase and production methods intensify and specialize, so the technologies of automated identification, feeding, cleaning and slaughter have greatly multiplied allowing economies of scale to retain or expand food chain profitability. From automatic milking systems, commonplace on dairy farms across the higher income countries, to automated water-bath stunning and neck-cutting systems in poultry slaughterhouses, automation―which we might simply define as processes that can take place rapidly and repetitively with minimal direct human action―is both a consequence, and a driving force, in contemporary livestock agriculture.

By definition, automation represents a reduction in the direct intervention of humans (farmers and stock-persons) in the daily lives of farmed animals. For many commentators automation is widely welcomed as liberating the farmer and stock-person from at least some of the repetitive, and often hard, labour of husbandry [1,2], thereby allowing gains in farmer well-being and reduced labour costs. However, as far back as in the early 1960s, the author and early critic of ‘factory farming’, Harrison identified automation as a problem of ‘unity’, arguing that:

*‘Mechanical cleaning reduces still further the time the stockman has to spend with them, and the sense of unity with his stock which characterizes the traditional farmer is condemned as being uneconomic and sentimental’*.([3], p. 35)

Recent analysis suggests that the current expansion of farm technologization requires new and often equally demanding data monitoring and management skills [4], and can also bring new forms of farm worker exploitation and marginalisation [5,6]. Nevertheless, the general spread of farm automation has been a key feature of contemporary agricultural systems and has had a major impact not only upon farm labour forces but also upon both the animal welfare monitoring roles they undertake and the impact of those roles upon the animals themselves.

Automation also impacts upon the nature and perception of farmed animal welfare. One might argue that automation helped indirectly to set the early parameters for defining the welfare of farmed animals. Accepting the essentially artificial (and ever more automated) nature of farming systems that emerged in the 1970s, welfare was initially defined normatively as the ability of animals to cope with these intensive environments [7]. For producers, a healthy animal has been essentially a productive one [8], with health—and indirectly, welfare—being measured according to standardised, and increasingly automated, performance indicators.

Nevertheless, automation, in reducing the levels of observation and frequency of contact between farmer and farmed animal, has been charged with potentially making that relationship, in Cornou’s words, ‘distant and impoverished’ [9], leading, on the one hand, to unrecognised or unacknowledged welfare issues [10,11] and, on the other, to a persistent oversimplification of welfare understanding [8]. As definitions and approaches to farmed animal welfare grow in both measurement and sophistication, the time farmers and stock-persons actually spend in observing the animals in their charge is held to constitute an important factor in defining and determining their welfare [12]. Hence, in 2006, the UK Farm Animal Welfare Council identified the ‘stockman’ as having: ‘a unique role within livestock farming in ensuring high standards of animal welfare’, further arguing, in an echo of Ruth Harrison, that; ‘the attributes of a good stockman include an affinity and empathy with livestock, patience, and keen observational skills amongst others’ [13]. Similarly, current European Union and national welfare rules charge farmers and stock-persons with the responsibility to inspect animals at regular intervals (usually, at least once a day) to verify their wellbeing [14], see also, for example, UK Government 2000; EU Council Directive 2008/119/EC [15]. Good welfare, it would seem, is in large part, dependent upon good observation.

In this paper, we argue that the current relationship between ‘good observation’ and ‘good welfare’ is no longer sufficient or appropriate. We note that despite an expanding raft of international and national animal welfare regulations, codes and practices, growing public awareness and an emerging sense of the marketable value of higher welfare products, there remain today considerable and persistent welfare concerns across all the major livestock production systems, ranging from the consistently high prevalence of lameness in dairy cattle and few possibilities for animals of all types to express natural behaviours in indoor systems. We ask, can Precision Livestock Farming (PLF) technologies address this observational hiatus?

Recent years have seen a series of major leaps forward in the technologies and methods of automated animal observation and monitoring, notably under the generic terminology of PLF whose general aim is to increase the efficiency of livestock farming systems and, in doing so, to take into account differences between individual animals or groups [16]. We argue that PLF technologies have a significant potential to support the management of animal welfare through radically new forms and mechanisms of observation. To date, however, that potential is both underdeveloped and under-studied. PLF technologies have the potential to monitor animal health and behaviour in ways that go beyond those of conventional welfare monitoring and observation. Moreover, they offer the opportunity to observe animal behaviour without interference.

In these ways, PLF technologies can help farmers detect problems and take management decisions to improve animal welfare at an early stage. The data derived from PLF technologies can be used to derive warnings and trigger notifications and alarms. Data can also be combined to create complex yet relevant welfare indicators that could be transmitted between food chain actors and developed into quality control procedures. Finally, PLF technologies offer the possibility of more holistic, continuous (whole-life) and standardised (comparable) welfare assessment for farmed animals across the full production chain. With this, of course, comes a certain number of concerns, not the least of them folding back to the role and expertise of the farmer/stock-person, now potentially an observer less of animals than of data, or to the established forms of agricultural advisory and decision-making structures that guide on-farm practice. As an increasing number of observers are pointing out, the growth of PLF technologies on farms raises new challenges for data management, distribution and ownership as well as recasting the role of the farmer or stock-person both with respect to the animals under their charge and to the valuable and sensitive data that increasingly represent them [17,18,19].

In this paper, we examine how PLF technologies and broader developments in information and communication technologies might improve the observation, monitoring and assessment of farmed animal welfare, particularly in those areas and systems where generic welfare problems persist. We consider how PLF might enhance our understanding of welfare, and the means to assess it.

## 2. The Promise of PLF 

The International Society for Precision Agriculture, established in the early 1990s to promote this area, produced an updated definition for Precision Agriculture:

*‘Precision Agriculture is a management strategy that gathers, processes and analyzes temporal, spatial and individual data and combines it with other information to support management decisions according to estimated variability for improved resource use efficiency, productivity, quality, profitability and sustainability of agricultural production’*.[20]

Although ‘Precision Agriculture’ initially emerged within the arable and crop sector [21], the technologies of real-time monitoring, data management and decision support are increasingly being applied to modern livestock systems. Wathes describes it as follows:

*‘Precision livestock farming, PLF, is an embryonic technology that applies the principles of process engineering to livestock farming. PLF requires a sensing system for inputs and outputs; a mathematical model of input/output relationships; a target and trajectory for controlled processes; and a model-based controller with actuators for process inputs. PLF has great potential to transform livestock production by efficient utilisation of nutrients, early warning of ill health, and reduction in pollutant emissions’*.[22]

However, PLF needs to be seen as more than simply better and more technologically advanced parameter monitoring and resource planning. For in its approach both to data and to the animal, PLF offers the potential for a more radical, almost ontological, reassessment of what welfare is and how it might be assessed. First, PLF, through the technologies of observation and monitoring, brings individual animals to the fore:

*‘A starting point in PLF is the recognition that each individual animal is […] a CIT [complex, individual, time-variant] system. This contrasts with more classical approaches where animals are considered as ‘an average of a population and due to its complexity as a steady state system’*.[23]

Second, PLF allows for the establishment of welfare indicators and triggers that are not dependent solely upon periodic human observation and measurement [24].

Over the last 10 years, PLF techniques have been adopted across a range of different livestock systems, specially the dairy sector [25,26]. The continually expanding demand for animal products, consumer (and retail) demand for more welfare friendly production, economic pressures and, at the same time, the need to reduce environmental impacts, all combine to require livestock farming to become ever-more efficient, leading to larger farms, labour reduction and increased mechanisation. PLF is seen as a major contributor to contemporary and efficient animal production; for example, by detecting oestrus prior to insemination, by adapting feeding to individual rates of animal weight gain and through the early detection of disease. Greater ‘efficiency’, whether in animal production, in resource and labour use, in feed and medication usage, in animal reproduction and so on, is the watchword of PLF. Some 35 of the 121 papers given at the EC-PLF Conference in 2019 explicitly addressed efficiency gains of PLF technologies [27]. Although still in relatively early stages of development, recent evidence shows that farmers and food chain actors are acquiring PLF technologies for a variety of reasons including the facilitation of their management roles and the improvement of their own well-being through reducing the workload especially for large herds and limiting repetitive tasks [28,29].

In practical terms, PLF technologies generally involve the fixing of sensors, or other forms of observational/monitoring technology, to individual animals and/or their environment, which continuously monitor a series of different parameters. Algorithms are then deployed to generate indicators and detect anomalies in these time-series data. A growing number of PLF technologies are now available on the market. These include: Real-Time Locating Systems (RTLS) to detect the position of animals and infer their activity (e.g., CowView (http://www.gea-cowview.com/), CowManager (https://www.cowmanager.com/en-us/));Accelerometers to measure whether an animal is standing, lying, moving and even eating or ruminating (e.g., Heat’Live, Time’Live and Feed’Live (https://www.cowmanager.com/en-us/), IceQube (https://www.icerobotics.com/researchers/));Cameras coupled with image analysis (e.g., Kinect cameras to detect aggression in pigs [30], RO-MAIN smaRt Cam (http://www.ro-main.com/en/products/details_products.php?no_produit=54) and EyeNamic (https://www.fancom.com/solutions/biometrics/eyenamic-behaviour-monitor-for-broilers) to describe the distribution and activity of pigs and poultry respectively, or a solution produced by Meyn to inspect footpads in poultry in the slaughterhouse);Sound recording to detect coughing or vocalisations from animals (e.g., SoundTalks (https://www.soundtalks.com/));Temperature and humidity recording, e.g., inside the vehicles that transport animals [31], or temperature of the animals themselves (e.g., Moow rumen bolus (http://moow.farm/));Weighing scales to weigh animals or control their feeding;Specific sensors to monitor biomarkers such as ruminal pH in cows (e.g., e-Cow FarmBolus (https://ecow.co.uk/the-ebolus-for-researchers/)), hormones (e.g., Herd Navigator with progesterone detection in milk) or gases (e.g., ChickenBoy (https://faromatics.com/products/));Electronic identification of large animals (cattle, pigs) thanks to Radio Frequency Identification (RFID) technology, allowing the tracking of animals in the various processing locations they may pass through.

The adoption of these technologies varies considerably. RFID and accelerometer technologies are well integrated, but other technologies still have to achieve a viable market share. As we have said above, most of the PLF technologies currently in commercial use are aimed principally at improving productive efficiency either through the close monitoring of such parameters as feed and oestrus or through the surveillance of (individual) animal health data. Indeed, we would argue that the primacy (and growing abundance) of productivity and health data through PLF technologies has to some extent distracted attention from both the continuing need for, and the possibilities of, data and PLF generated indicators of animal welfare. Moreover, where they have been applied, such technologies have tended to focus primarily on conventional production systems, with less attention given to systems that place a higher premium of animal welfare [32]. Hence, although commentators such as Berckmans [24] and Blokhuis [33]), strongly advocate the use of PLF to monitor animal behaviour, and thereby to address welfare issues in broilers and in pigs, both acknowledge that considerable challenges and trade-offs remain.

## 3. The Potential of PLF to Monitor Animal Welfare

With growing societal awareness and concern for the welfare of farmed animals, as evidenced in countless reports and publications as well as a growing raft of legislation and welfare certification, PLF technologies face a particular quandary. Inevitably and inescapably associated with the increasing automation, technologisation and industrialisation of livestock farming (though perhaps ironically, offering enhanced possibilities to closely monitor the states of individual animals), PLF needs to demonstrate its relevance and contribution to improving the welfare of farmed animals. As we have said elsewhere, PLF technologies are likely to come under significant pressure, both from food chain actors and consumers, to provide valid welfare gains [34,35]. Some PLF companies certainly have begun to explicitly address welfare issues through their particular technologies (such as the specifically welfare oriented Time’Live tool). The Chickenboy, for example, has described its technology as an ‘autonomous, smart broiler robot that monitors air quality, health and welfare and equipment operation’ while eYeNamic offers ‘valuable indicators of broiler welfare’. The challenge is that welfare is now recognized as a complex physiological and psychological state, comprised of multiple dimensions. To date, PLF technologies have focused primarily upon the physical components of health and welfare (growth, feeding, and health). The challenge for PLF, from an animal welfare perspective, is to combine and assemble data of potentially widely differing types and varied sources into meaningful indicators that include the other aspects of welfare, such as mental state, naturalness and resilience.

It is axiomatic that European livestock farming still faces a number of major and persistent and system-wide animal welfare problems [36,37,38,39,40] such as lameness in dairy cattle, tail biting in pigs, poor locomotion in broilers as well as long-standing generic issues in transport and slaughter. Moreover, the active promotion of positive welfare states and the encouragement of better opportunities for animals to express natural behaviour, the value of which in welfare terms are widely acknowledged by science [41,42,43] and valued by consumers [44], remain largely incompatible (or unmeasurable) with many contemporary production systems. If it is to make a substantive contribution to addressing genuine animal welfare concerns, PLF technology must therefore address itself to these two critical areas; the effective monitoring and identification of systemic welfare failures and the active enhancement of opportunities for positive welfare experiences. In this, however, PLF technologies need to go further than the ad hoc or corrective addressing of specific and persistent welfare problems. Ultimately, PLF technologies need to be integrated, from the start, into the design of new farming systems and production technologies [45] so that animal welfare becomes an integral component of them. 

The Welfare Quality Project identified 12 key criteria that need to be addressed to cover all aspects of animal welfare [46]. Innovative applications of the various PLF approaches mentioned above, along with others still in development, could, we argue, generate crucial data for the effective monitoring of all of these, as Molina et al. [47] have recently demonstrated with respect to dairy cows. Current PLF technologies already offer the opportunity to provide robust and continuous information across a range of measures identified by the Welfare Quality research as listed in Table 1. 

The approaches to defined outcomes in Table 1 are predominantly associated with welfare on farm. The monitoring of ambient conditions and the recording of physiological parameters on animals (such as heart and respiratory rates) could help to assure that animals are transported in appropriate conditions. Similarly, at slaughter, PLF technologies that allow for the monitoring and analysis of animal movement to ensure effective welfare at slaughter are currently being investigated [60].

For many of these welfare concerns, the PLF observational and monitoring technology is available. However, what is required are appropriate algorithms and data assemblages to enable effective welfare indicators to be developed and to determine mechanisms for intervention when deviations from normal or intended positive states are recorded. Such indicators, we maintain, will require that data from different sensors, are combined with other data sources and interpreted together. Thus, for example an effective indicator of dairy cow welfare would need to bring together individualised data on heart and respiratory rates, (which will vary with days in lactation), milk yield, ambient parameters that assess heat load as well as environmental and behavioural data [61,62].

As PLF technologies and tools develop, we estimate that they will have significant potential to increase our capacity to monitor and address animal welfare. They will provide the means to monitor farmed animal health and welfare and detect disorders thereby enabling both rapid and accurate treatment of disease or behavioural disorders and the prevention of their spread to other animals. Moreover, PLF technologies will facilitate the assessment of animal premises and farm environments (including transport and slaughter environments). For both, PLF technologies allow for continuous, rather than ‘snap-shot’ observation notably across periods and in locations where human observation is either problematic or impossible.

It seems to us essential that technology developers, biologists, ethologists and veterinarians work closely together in developing the opportunities for new observational and monitoring tools that specifically address welfare issues at different points in the livestock production chain, from farm to slaughterhouse. Critically, this will require the design of appropriate algorithms, data trigger points, alerts and warning systems, along with other mechanisms that allow for individual and generic welfare incidents and issues to be identified and subsequently addressed by producers. 

## 4. Perspectives to Improve Animal Welfare in a Digital World 

Livestock farming is currently experiencing something of a numerical revolution with the multiplication of environmental and animal health sensors and their integration into various large data platforms and information systems. To date, the primary objectives of this new observational technology have been animal health and productivity. However, there is also an imperative that these new approaches be employed both in the monitoring and improvement of other aspects of farmed animal welfare and in the repositioning of welfare in the design of new and innovative production systems. Research and innovation is particularly needed in the development of, first, effective data management systems to allow for effective welfare indicators and alerts to be constructed and relayed to farm (or transport and slaughterhouse) personnel and, second, efficient data storage and recording techniques to permit the construction of relevant time-series data that can inform processes of benchmarking, certification and compliance monitoring. Both of these offer the potential to take farm animal welfare monitoring and, ultimately improvement, forward in new, dynamic and innovative ways. As such, animal welfare scientists, ethologists and veterinarians have a critical input to make to these necessary processes. However, both also raise certain challenges in themselves. As Klerkx et al. [19] pointed out, relatively little attention has been paid so far within the precision farming literature to the complex societal questions around data use, ownership and circulation and how different food chain actors, from food processors and retailers to veterinarians and farmers, might use such technologies and data to ultimately, in their words, transform agricultural production systems, value chains and food systems.

We recognise that the growth of PLF technology and its specific application to the monitoring and improvement of farmed animal welfare raises a number of broader issues that also need to be addressed. First amongst these are the complex technical and scientific issues around welfare definition, the establishment of effective welfare indicators and the validity of the data collected by the new technologies of observation and surveillance. The production of welfare indicators has been an important area of recent animal welfare science and increasingly indicator protocols are being used by commercial food chain actors both to segment markets and to reassure concerned consumers. While PLF can provide new and complex data on animal health and behaviour, the development of relevant algorithms to combine individual data sets into meaningful indicators remains a significant challenge. How that data, and combinations thereof, are translated into cues for action and intervention, is an associated technical concern that will necessitate not only effective alert systems but also relevant training for farm personnel.

A second issue is the perceived difficulties of applying PLF technologies to extensive livestock systems. To date, PLF technologies have been primarily targeted at intensive, high input, high technology livestock systems, where animals are often housed in relative high densities. While this has undoubtedly been the dominant model of global agricultural development over the last few decades, less intensive and more ‘sustainable’ forms of livestock farming, and specific support mechanisms for their extension, have also seen a significant rise in recent years, particularly within Europe. However, extensive livestock systems have not seen the rapid adoption of precision farming technologies that has characterized intensive production [63] for a number of complex structural, environmental and technological reasons [64]. Nevertheless, while, we note a growing research interest in extending PLF to extensively raised animals through such approaches as group or herd monitoring, GPS and RFID technologies [65,66,67], such approaches, like their equivalent in intensive systems, have tended to date to focus predominantly upon productivity and health rather than welfare.

A third issue relates to the nature, deployment and ownership of the data produced by PLF technologies [18,68]. Currently, most of the data produced by existing PLF technologies is site specific (i.e., it serves to adjust the management on a given farm, transport unit or slaughterhouse, and is not exchanged between the food chain actors). Moreover, it is rarely (if at all) aligned with other on-farm data sets, such as data from automatic milking systems, veterinary surveillance data or post-mortem data from slaughterhouses. The full potential of more integrated data fields is still a long way off [69], although many food chain actors are beginning to develop integrated data management systems across their supply chains for quality assurance purposes, aimed both at consumers and contracted producers. Examples of this include De Hoeve Innovatie (KDV) in the Netherlands and Carrefour in France who use sophisticated tracking technologies for their animal products. While, for the moment at least, such data sets are generally private, there would be major advantages if large-scale commercial monitoring and surveillance data was shared with producers and veterinarians for benchmarking, early warning of disease and behavioural issues, assessment of fitness for transport and so on. In short, new business models for the distribution and use of PLF monitoring data are required if greater transparency is to be achieved and if farm animal welfare, as a public good, is to be lastingly improved.

A final concern brings us back to the farmers and stock-persons with whom we began this article. Under PLF their roles and their expertise, in addition to their relationship to the animals under their care, changes and this too requires management and adaptation. The daily walk-through the broiler shed may well be replaced by the periodic reading of on-screen data or the reception of phone-alerts. As [9] points out, actual contacts with farmed animals may be reduced to physical interventions of a less welfare-friendly nature, thereby exacerbating the potential for negative animal responses to human presence. Thus, as the nature of husbandry work shifts, so too does the nature of animal care. On the other hand, when technologies relieve the workload, farmers can spend more time close to their animals, as was observed in some farms acquiring an automatic milking system [70]. Given the possible associations we identify above between PLF and consumer concerns for the automation of husbandry, this too needs to be carefully thought through. While some might argue that farmed animals are more likely to exhibit ‘natural’ behaviour when humans are absent, the environments in which they live are, to a greater or lesser extent, far from natural and our responsibility for their care no less strong.

## 5. Conclusions

PLF offers significant opportunities for a more holistic, evidence-based approach to the monitoring and surveillance of farmed animal welfare. To date, the emphasis within existing PLF technologies has been predominantly on animal health and productivity and, as such technologies develop further along these lines, we must ensure that these alone do not come to define welfare. To achieve this, PLF developers, animal welfare scientists and food chain actors need to collaborate, along with farmers and veterinarians, in widening the remit of PLF to address welfare issues in a more holistic and technologically creative manner. This needs to take place along three co-related fronts: first, in extending the capacity of PLF technologies to identify and measure appropriate welfare issues—suggesting the need for collaboration between welfare scientists and PLF technology developers; second, in addressing the complex issue of data construction and its translation into effective mechanisms for alert and response—suggesting an important role for on-farm demonstration and training; and third, in seeking ways of validating effective PLF welfare monitoring into quality assurance mechanisms and the value chain—implying an extended collaboration between welfare science, NGOs, food chain actors and consumer interests. In driving all of these forward, we believe that contemporary animal welfare science has a leading role to play. 

## Figures and Tables

**Table 1 animals-10-01779-t001:** Examples of possible PLF approaches to the Welfare Quality criteria of good welfare.

Welfare Outcome	PLF Approach
Absence of prolonged hunger	Body condition score of cattle assessed by imaging technologies [48], time spent waiting in front the feeding table when food is not available
Absence of prolonged thirst	Time spent at drinkers can be detected using RFID detectors [49] or camera tracking [50]
Comfort around resting	Time spent in lying areas can be obtained from RTLS technologies, the time spent actually lying down can be recorded with accelerometers
Thermal comfort	Sensors to measure heart rate or respiratory rates (which are increased in case of high heat load) already exist at least for large species; the ambient conditions are already monitored (temperature, humidity),
Ease of Movement	The use of the different areas can be assessed through RTLS with ultra wide band emitters inside or GPS outside
Absence of injuries	External injuries can be detected at least at slaughter from image analysis [51]
Absence of disease	Continuous monitoring of animals’ behaviour can detect changes in time budgets (time spent feeding, ruminating, resting, walking, etc.) and in circadian rhythms of activities and such behavioural changes may reflect sickness [52,53,54,55]. Monitoring of coughs can indicate respiratory disease [56]
Absence of pain induced by management procedures	Experience of pain can be detected from facial expressions in sheep [57] and pigs (ongoing project from Bristol Robotics Laboratory (https://www.bbc.com/news/av/science-environment-49362428/pigs-emotions-could-be-read-by-new-farming-technology))
Expression of social behaviours	The functioning of the social group can be on the basis of the animals’ interactions, positions and activity [30,58]
Expression of other behaviours	The use by farmed animals of specific welfare enhancing resources can be monitored. For example, the use of brushes by cattle can be detected by accelerometers on brushes coupled while the animals proximity to the brush can be detected with an RFID detector or a RTLS [59] the use of outdoor areas by poultry can be monitored using infrared beams that detect when the birds go through the passage to the outdoor area.

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
