# Peer review of "Animal Welfare Management in a Digital World"

_animals, 2020, doi:10.3390/ani10101779_

Round 1
Reviewer 1 Report
The topic of the paper is of huge interest and the Authors addressed it presenting a very clear analysis and discussion.
In my opinion the paper is worth of been published without the need of changes except for a check of the references.
In particular, please complete/correct the following references:
n° 16 Guarino, M.; Berckmans, D. Precision Livestock Farming '15. In Proceedings of EC-PLF, Milano.
n° 22 Wathes, C.M. Precision livestock farming for animal health, welfare and production. In Proceedings of roceedings of the ISAH 2007; pp. 397-404.
n° 23 Berckmans, D. Automatic On-line Monitoring of animal Health and welfare by Precision livestock farming. In Proceedings of International Society for Animal Hygiene (ISAH) Saint Malo, France; pp. 27-30.
n° 51 Van Harn, J.; De Jong, I.C. Validation Meyn Footpad Inspection System; 2017.
n° 57 Lu, Y.; Mahmoud, M.; Robinson, P. Estimating sheep pain level using facial action unit detection. In Proceedings of IEEE International Conference on Automatic Face and Gesture Recognition, 30 May - 3 June, 2017, Washington, DC.
n° 58 Rocha, L.E.C.; Veissier, I.; Terenius, O.; Meunier, B.; Nielsen, P.P. Real-Time Locating System to study the persistence of sociality in large-mammal group dynamics. In Proceedings of EC-PLF, Cork, 26-29 September 2019.
n° 65 Kling-Eveillard, F.; Hostiou, N. The effects of PLF on human animal relationship on farm. In Proceedings of 8th European Conference on Precision Livestock Farming, ECPLF, Nantes, France; p. 9 pp.
Author Response
We thank the reviewers for noticing errors and for their valuable comments; We amended our manuscript accordingly. Below are explanations on the changes we made.
We check the references highlighted by Reviewer 1 and corrected them as follows:
n° 16 Guarino, M.; Berckmans, D. Precision Livestock Farming '15. In Proceedings of EC-PLF, Milano.
Now: Guarino, M.; Berckmans, D. Precision Livestock Farming '15; Milano univ.: Milano, Italy, 2015; pp. 868.
n° 22 Wathes, C.M. Precision livestock farming for animal health, welfare and production. In Proceedings of roceedings of the ISAH 2007; pp. 397-404.
Now: Wathes, C.M. Precision livestock farming for animal health, welfare and production. In Proceedings of XIII Iinternational congress in animal hygiene -ISAH 2007, Tartu, Estonia; pp. 397-404.
n° 23 Berckmans, D. Automatic On-line Monitoring of animal Health and welfare by Precision livestock farming. In Proceedings of International Society for Animal Hygiene (ISAH) Saint Malo, France; pp. 27-30.
Now: Berckmans, D. Automatic on-line monitoring of animal health and welfare by precision livestock farming. In Proceedings of X International congress in animal hygiene -ISAH 2004, Saint-Malo, France; pp. 27-30.
n° 51 Van Harn, J.; De Jong, I.C. Validation Meyn Footpad Inspection System; 2017.
Now: Van Harn, J.; De Jong, I.C. Validation Meyn footpad inspection system; Wageningen Livestock Research Report 1044B, Wageningen, The Netherlands, 2017.
n° 57 Lu, Y.; Mahmoud, M.; Robinson, P. Estimating sheep pain level using facial action unit detection. In Proceedings of IEEE International Conference on Automatic Face and Gesture Recognition, 30 May - 3 June, 2017, Washington, DC.
Sorry there seem to be no other specifications except DOI à we added the DOI (10.1109/Fg.2017.56).
n° 58 Rocha, L.E.C.; Veissier, I.; Terenius, O.; Meunier, B.; Nielsen, P.P. Real-Time Locating System to study the persistence of sociality in large-mammal group dynamics. In Proceedings of EC-PLF, Cork, 26-29 September 2019.
These results are now published as an article in appli anim behave sci à we replaced the former citation by that of the published article:
Rocha, L.E.C.; Terenius, O.; Veissier, I.; Meunier, B.; Nielsen, P.P. Persistence of sociality in group dynamics of dairy cattle. Appl. Anim. Behav. Sci. 2020, 223, 104921, doi:10.1016/j.applanim.2019.104921.
n° 65 Kling-Eveillard, F.; Hostiou, N. The effects of PLF on human animal relationship on farm. In Proceedings of 8th European Conference on Precision Livestock Farming, ECPLF, Nantes, France; p. 9 pp.
Now: Kling-Eveillard, F.; Hostiou, N. The effects of PLF on human animal relationship on farm. In Proceedings of Precision Livestock Farming '17, Nantes, France; pp. 725-736.
Reviewer 2 Report
Review: Animal welfare management in a digital world
The authors have clearly identified the importance of not neglecting a clear focus on welfare in inputs and analysis of data gathered in PLF. The challenged is that health can often be inferred directly from such data, such as milk cell count or body temperature, but that welfare often has to be secondarily derived form outcome measures previously shown to be linked to welfare states.
The authors conclude that what is needed is a “..combined multi-actor approach that brings together industry, scientists, food chain actors, policy-makers and NGOs to develop and use the promise of PLF for the creative and effective improvement of farmed animal welfare. “
This is a laudable aim, but would benefit from some consideration of how this might proceed. I think they have identified this in lines 297-2999: “ How that data, and combinations thereof, are translated into cues for action and intervention, is an associated technical concern that will necessitate not only effective alert systems but also relevant training for farm personnel.”
So I would suggest that their conclusion is built on from “something must be done…” to “..and this is how we suggest that something could be started”.
Whilst I also agree that intensification is an important worldwide trend, so is sustainable agriculture with lower inputs. Professor Buller will be aware of changes in farm support systems in the UK with the new England/Wales Agriculture Bill, such that more sustainable extensive grassland beef production may be more supported than systems with crop feed inputs, raised a further challenge. How can PLF, and in term its measurement of welfare outcomes in these extensive situations, be developed? A mention of these challenges could be useful.
A small, but for me, important point from lines (L) 49-54. Whilst it is inherently correct to say “automation represents a reduction in the direct intervention of humans” it is of concern that Harrison’s comment is introduced first “‘Mechanical cleaning reduces still further the time the stockman has to spend with them, and the sense of unity with his stock which characterizes the traditional farmer is condemned as being uneconomic and sentimental’ (p. 35). “It could be less indicative of possible perception of bias if this concept were introduced after Lines 55-58, the which could righty be postulated as the accepted view ( of many), as a reasonable contrast:
“By definition, automation represents a reduction in the direct intervention of humans (farmers and stock persons) in the daily lives of farmed animals. For many commentators, automation is widely welcomed as liberating the farmer and stock-person from at least some of the repetitive, and often hard, labour of husbandry [2,3], thereby allowing gains in farmer well-being and reduced labour costs. However as far back in the early 1960s, the author and early critic of ‘factory farming’, Harrison [1] identified automation as a problem of ‘unity’: ‘Mechanical cleaning reduces still further the time the stockman has to spend with them, and the sense of unity with his stock which characterizes the traditional farmer is condemned as being uneconomic and sentimental’ (p. 35). Recent analysis also suggests …”
Overall a useful paper that provides an important challenge as PLF develops.
Author Response
The authors have clearly identified the importance of not neglecting a clear focus on welfare in inputs and analysis of data gathered in PLF. The challenged is that health can often be inferred directly from such data, such as milk cell count or body temperature, but that welfare often has to be secondarily derived form outcome measures previously shown to be linked to welfare states.
The authors conclude that what is needed is a “..combined multi-actor approach that brings together industry, scientists, food chain actors, policy-makers and NGOs to develop and use the promise of PLF for the creative and effective improvement of farmed animal welfare. “
This is a laudable aim, but would benefit from some consideration of how this might proceed. I think they have identified this in lines 297-2999: “ How that data, and combinations thereof, are translated into cues for action and intervention, is an associated technical concern that will necessitate not only effective alert systems but also relevant training for farm personnel.”
So I would suggest that their conclusion is built on from “something must be done…” to “..and this is how we suggest that something could be started”.
AU: We rewrote this part of the discussion suggesting avenues to be investigated to addresse end of the conclusion as follows (Lines 348-361):
To achieve this, PLF developers, animal welfare scientists and food chain actors need to collaborate, along with farmers and veterinarians, in widening the remit of PLF to address welfare issues in a more holistic and technologically creative manner. This needs to take place along three co-related fronts: first, in extending the capacity of PLF technologies to identify and measure appropriate welfare issues – suggesting the need for collaboration between welfare scientists and PLF technology developers; second, in addressing the complex issue of data construction and its translation into effective mechanisms for alert and response – suggesting an important role for on-farm demonstration and training; and third, in seeking ways of validating effective PLF welfare monitoring into quality assurance mechanisms and the value chain – implying an extended collaboration between welfare science, NGOs, food chain actors and consumer interests. In driving all of these forward, we believe that contemporary animal welfare science has a leading role to play.
Whilst I also agree that intensification is an important worldwide trend, so is sustainable agriculture with lower inputs. Professor Buller will be aware of changes in farm support systems in the UK with the new England/Wales Agriculture Bill, such that more sustainable extensive grassland beef production may be more supported than systems with crop feed inputs, raised a further challenge. How can PLF, and in term its measurement of welfare outcomes in these extensive situations, be developed? A mention of these challenges could be useful.
AU: We added a paragraph to discuss the use of PLF in extensive system this point (Lines 300-312):
A second issue is the perceived difficulties of applying PLF technologies to extensive livestock systems. To date, PLF technologies have been primarily targeted at intensive, high input, high technology livestock systems, where animals are often housed in relative high densities. While this has undoubtedly been the dominant model of global agricultural development over the last few decades, less intensive and more ‘sustainable’ forms of livestock farming, and specific support mechanisms for their extension, have also seen a significant rise in recent years, particularly within Europe. However, extensive livestock systems have not seen the rapid adoption of precision farming technologies that has characterized intensive production [63] for a number of complex structural, environmental and technological reasons [64]. Nevertheless, while, we note a growing research interest in extending PLF to extensively raised animals through such approaches as group or herd monitoring, GPS and RFID technologies [65-67], such approaches, like their equivalent in intensive systems, have tended to date to focus predominantly upon productivity and health rather than welfare.
A small, but for me, important point from lines (L) 49-54. Whilst it is inherently correct to say “automation represents a reduction in the direct intervention of humans” it is of concern that Harrison’s comment is introduced first “‘Mechanical cleaning reduces still further the time the stockman has to spend with them, and the sense of unity with his stock which characterizes the traditional farmer is condemned as being uneconomic and sentimental’ (p. 35). “It could be less indicative of possible perception of bias if this concept were introduced after Lines 55-58, the which could righty be postulated as the accepted view ( of many), as a reasonable contrast
“By definition, automation represents a reduction in the direct intervention of humans (farmers and stock persons) in the daily lives of farmed animals. For many commentators, automation is widely welcomed as liberating the farmer and stock-person from at least some of the repetitive, and often hard, labour of husbandry [2,3], thereby allowing gains in farmer well-being and reduced labour costs. However as far back in the early 1960s, the author and early critic of ‘factory farming’, Harrison [1] identified automation as a problem of ‘unity’: ‘Mechanical cleaning reduces still further the time the stockman has to spend with them, and the sense of unity with his stock which characterizes the traditional farmer is condemned as being uneconomic and sentimental’ (p. 35). Recent analysis also suggests …”
AU: We changed the text accordingly (Lines 49-57):
By definition, automation represents a reduction in the direct intervention of humans (farmers and stock-persons) in the daily lives of farmed animals. For many commentators automation is widely welcomed as liberating the farmer and stock-person from at least some of the repetitive, and often hard, labour of husbandry [1,2], thereby allowing gains in farmer well-being and reduced labour costs. However, as far back as in the early 1960s, the author and early critic of ‘factory farming’, Harrison [3] identified automation as a problem of ‘unity’, arguing that:
‘Mechanical cleaning reduces still further the time the stockman has to spend with them, and the sense of unity with his stock which characterizes the traditional farmer is condemned as being uneconomic and sentimental’ (p. 35).
Overall a useful paper that provides an important challenge as PLF develops.